# Sexual Orientation and Gender Identity Associated with Sexual Practices, Psychoactive Substance Use and Sexually Transmitted Infections Among HIV PrEP Users

**DOI:** 10.3390/healthcare13151841

**Published:** 2025-07-29

**Authors:** Marcos Morais Santos Silva, Lucas Cardoso dos Santos, Mayara Maria Souza de Almeida, Lucia Yasuko Izumi Nichiata

**Affiliations:** Nursing School, University of São Paulo, São Paulo 05403-00, Brazil; lucas.csantos@usp.br (L.C.d.S.); mayaramaria@usp.br (M.M.S.d.A.);

**Keywords:** pre-exposure prophylaxis, HIV, sexual health

## Abstract

HIV disproportionately affects key populations (MSM, transgender people, sex workers and psychoactive substance users), who face greater social vulnerability and limited healthcare access. This study aimed at analyzing sexual orientation and gender identity and their association with sexual practices, sexually transmitted infections and psychoactive substance use among PrEP users. **Method:** A cross-sectional study was conducted between January 2018 and June 2021 with 736 Brazilian PrEP users from a health service in São Paulo. Sociodemographic data, sexual behaviors, STI history (past 3 months) and psychoactive substances use (past 3 months) were extracted from clinical records. The associations were analyzed using binomial logistic regression (*p* < 0.05). **Results:** Most of the participants were cisgender men (93.4%) and homosexual (84.8%), with a mean age of 34.9 years old. Condomless sex was reported by 98.5%, and 18.4% had some recent sexually transmitted infection, mainly syphilis. Psychoactive substance use was reported by 55.4%, especially marijuana, club drugs, erectile stimulants and poppers. Transgender and cisgender women were more likely to report sex work and crack use. Homosexual and bisexual participants had higher odds of using erectile stimulants. **Conclusions:** The study reveals key links between gender, sexual orientation and risk behaviors, highlighting the need for inclusive, targeted prevention.

## 1. Introduction

Despite advances in the efficacy of and access to antiretroviral drugs (ART), human immunodeficiency virus (HIV) remains one of the main global public health challenges, especially in contexts marked by limited resources, regional disparities and more vulnerable populations [1]. In 2023, nearly 39.9 million people were living with HIV and approximately 630,000 deaths were attributed to AIDS-related diseases, representing a global prevalence of 0.8% among adults [2].

The HIV epidemic is more concentrated in certain groups. Its prevalence among transgender women can reach 24.5% in Latin America [3], while the rates vary from 2.4% to 29% in European countries and reach around 11.4% in Latin America [4] among men who have sex with men (MSM). In Brazil, the epidemic remains concentrated among MSM, transgender people, sex workers and psychoactive substance (PAS) users [5].

Transgender people, gays, bisexuals and other identities belonging to the LGBT community face barriers in accessing health services, in addition to suffering institutional discrimination and stigmatization, which increases exposure to HIV and reduces adherence to preventive measures [6]. Presence of these vulnerabilities is aggravated by contexts marked by social exclusion and violence, which makes these groups more susceptible to HIV infection [7].

In this context of vulnerability, practices such as chemsex are also observed, involving the intentional use of psychoactive substances to prolong and intensify sexual experiences. This behavior is associated with an increase in the number of sexual partners and a reduction in condom use [8], which raises the likelihood of HIV transmission and other sexually transmitted infections (STIs) [8,9], such as syphilis, gonorrhea, chlamydia and hepatitis C.

In this scenario, pre-exposure prophylaxis (PrEP) stands out as an effective strategy for HIV prevention. It consists of the continuous administration of antiretroviral drugs to individuals with an increased chance of exposure. It has demonstrated high efficacy in clinical trials, with reductions of up to 99% in HIV incidence when properly used [10,11]. In Brazil, PrEP was incorporated in 2017 into the Brazilian Unified Health System (*Sistema Único de Saúde*, SUS), a universal health system, initially offered in specialized services and was also made available in Primary Health Care services [12] from 2021 onwards, which expanded access to combined HIV prevention.

Despite the benefits of PrEP, questions arise as to the possible relationship between sexual practices associated with an increase in the incidence of STIs among its users. Furthermore, studies investigating which social groups access PrEP more frequently and how PAS use, gender identity and sexual orientation are associated with the occurrence of STIs and risky sexual practices are still limited [13,14]. Understanding these aspects is essential in order to strengthen prevention actions and increase the effectiveness of public policies aimed at sexual and reproductive health.

The relevance of this research lies in supporting evidence-based interventions that are sensitive to the particularities of the most vulnerable populations, in order to help improve combined prevention strategies and promote comprehensive and equitable care. Therefore, this study aims to analyze how sexual orientation and gender identity are associated with sexual practices, psychoactive substance use and the occurrence of sexually transmitted infections among users of HIV pre-exposure prophylaxis (PrEP).

## 2. Materials and Methods

This cross-sectional study was carried out at a health service specialized in STIs/AIDS located in the capital city of the state of São Paulo, Brazil, the country’s largest metropolis, which provides care for nearly 7000 users a month through consultations, group care appointments, laboratory tests and rapid testing for HIV, syphilis and hepatitis B and C, application of medicines and vaccines, dispensation of pre- and post-exposure prophylaxis and treatment for STIs/AIDS, among others.

The service employs protocols to systematize care for HIV PrEP users. At the first visit, each user is registered for the program with information on sociodemographic characteristics, the reason for seeking the service and personal history. At the return visits (30, 60, 90 and 120 days after the first one), the aim is to assess adherence, discontinuation, medication side effects, alcohol and other drug use, sexual practices, as well as HIV diagnosis and seroconversion among people using PrEP.

Although adherence, discontinuation and seroconversion were not analyzed in this study, some variables related to sexual practices and psychoactive substance use (e.g., alcohol, marijuana, poppers, club drugs, erectile stimulants and solvents) were collected during the first follow-up visit (30 days after PrEP initiation), as per the standardized protocol of the service.

The study participants were service users who started using PrEP between January 2018 and December 2021. The inclusion criteria were as follows: agreeing to take part in this research, being users of the specialized service in question, being at least 18 years old and using HIV PrEP for the first time.

A standardized instrument was used to extract data from users’ electronic medical records, including the independent variables. The dependent variables referred to sexual practices, history of sexually transmitted infections (STIs) and psychoactive substance use. The data were analyzed using R Studio, version 4.1.1. Some variables presented a higher proportion of missing data. However, no imputation or aggregation of categories was performed. Given the underrepresentation of certain gender and sexual minority groups in research, we chose to preserve the integrity of the available data without modifying or excluding these participants, despite the limitations in statistical power.

Absolute and relative numbers were employed for the descriptive analysis, with simple frequencies for the qualitative variables. Central tendency (mean and median) and dispersion (range and standard deviation) measures were calculated for the quantitative variables. Associations between categorical variables were assessed using Pearson’s Chi-square test or Fisher’s exact test, as appropriate. Univariate binary logistic regressions were then performed to estimate the association between gender identity/sexual orientation and the outcomes of interest. A 95% confidence interval and 5% significance level were adopted.

## 3. Results

Table 1 presents data from 736 PrEP users, with a mean age of 34.9 years (SD = 9.1). There were no significant differences in age by gender identity (*p* = 0.506): cisgender men (*n* = 693; 94.1%) had a mean age of 34.8 years, cisgender women (*n* = 21; 2.8%) 36.2 years, and transgender women (*n* = 22; 3.0%) 34.1 years. However, age differed significantly by sexual orientation (*p* = 0.020): bisexuals (*n* = 76; 10.3%) had a lower mean age of 32.6 years (SD = 9.8), compared to homosexuals/gays/lesbians (*n* = 623; 84.7%) with 35.1 years (SD = 8.9) and heterosexuals (*n* = 33; 4.5%) with 35.4 years (SD = 11.3).

Most participants were cisgender men (*n* = 693; 94.1%), white-skinned (*n* = 494; 71.3%) and had ≥12 years of schooling (*n* = 595; 85.9%). Among cisgender women, 11 (52.4%) were white and 9 (42.9%) were brown-skinned, while among transgender women, 14 (63.6%) were brown-skinned and 6 (27.3%) were white. Educational level was lower among transgender women: 5 (22.7%) had 4–7 years, 9 (40.9%) had 8–11 years, and only 8 (36.4%) had ≥12 years of education (*p* < 0.001). Among heterosexuals, 17 (51.5%) were brown-skinned and only 19 (57.6%) had ≥12 years of education, while 546 (87.1%) of homosexuals/gays/lesbians had ≥12 years of education (*p* < 0.001).

Regarding sexual practices, 170 (24.7%) of cisgender men reported having sex with people living with HIV. Among cisgender women (*n* = 21), 11 (52.4%) reported not knowing their partners’ serological status, and the same was true for 39 (52.0%) of bisexual individuals. Sex in exchange for money was more frequent among transgender women (*n* = 16; 72.7%) and heterosexuals (*n* = 21; 63.6%), with significant differences by gender identity and sexual orientation (*p* = 0.001 for both).

Most participants reported condomless sex: 605 (98.9%) cisgender men, 18 (90.0%) cisgender women, 17 (94.4%) transgender women, 546 (98.6%) homosexuals, 67 (98.5%) bisexuals and 27 (96.4%) heterosexuals. These differences were statistically significant only across gender identity (*p* = 0.017). As for STI history, syphilis diagnosis in the last three months was reported by 65 (50.4%) of cisgender men and 3 (100%) of transgender women. Rectal gonorrhoea and/or chlamydia occurred in 29 (22.5%) of cisgender men and 5 (25.0%) of bisexuals, with no significant differences between groups (*p* > 0.05).

More than half of the participants reported psychoactive substance use, with no statistically significant differences by gender identity (*p* = 0.838) or sexual orientation (*p* = 0.716). Among cisgender men, the most frequently reported substances were cannabis (*n* = 233; 62.0%), club drugs (*n* = 164; 43.6%) and erectile stimulants (*n* = 159; 42.3%). Among transgender women, cannabis (*n* = 9; 69.2%) and cocaine (*n* = 7; 53.9%) were the most used. Heterosexual individuals reported high use of cocaine (*n* = 10; 52.6%) and cannabis (*n* = 13; 68.4%), but no use of erectile stimulants (*n* = 0; 0.0%). Statistically significant differences were observed by gender identity for crack use (*n* = 2; 9.1%; *p* = 0.041) and by sexual orientation for crack (*p* = 0.048), cocaine (*p* = 0.050), solvents (*p* = 0.038) and erectile stimulants (*p* = 0.001).

The univariate analysis assessing the association between gender identity, sexual orientation and the occurrence of sexually transmitted infections (STIs). Among the key findings, cisgender women (OR = 9.8; 95% CI: 3.5–27.3) and transgender women (OR = 15.6; 95% CI: 5.3–46.1) showed significantly higher odds of reporting exchanging sex for money compared to cisgender men. Cisgender women also reported lower odds of condomless sex (OR = 0.1; 95% CI: 0.0–0.5), while both cisgender (OR = 13.1; 95% CI: 1.31–130) and transgender women (OR = 9.9; 95% CI: 1.0–94.7) had notably increased odds of crack use.

With respect to sexual orientation, homosexual/gay/lesbian participants were less likely to report sex work (OR = 0.1; 95% CI: 0.0–0.2) and condomless sex (OR = 0.1; 95% CI: 0.0–0.6). Bisexual individuals also showed reduced odds of engaging in sex work (OR = 0.4; 95% CI: 0.2–1.0). Importantly, the odds of reporting a syphilis diagnosis were substantially higher among homosexual/gay/lesbian (OR = 29.0; 95% CI: 1.1–762) and bisexual individuals (OR = 34.6; 95% CI: 1.2–960) compared to heterosexual participants (Table 2).

## 4. Discussion

The results can be summarized as follows: cisgender, white-skinned men with more than 12 years of study were the ones who used prophylaxis the most. Transgender women were more likely to report exchanging sex for money, while cisgender and homosexual men reported higher frequencies of condomless sex, in contrast to cisgender women, who were less likely to engage in unprotected sex. Homosexual/gay/lesbian and bisexual individuals were also less likely to report sex work and condomless sex. However, both groups showed a greater likelihood of reporting a syphilis diagnosis when compared to heterosexuals. Regarding psychoactive substance use, marijuana was more frequently reported by cisgender men, cisgender women and heterosexuals, while crack use was more prevalent among cisgender and transgender women.

The sociodemographic profile of HIV PrEP users detected in this study is similar to other findings in the literature, which suggest that they are predominantly Men who have Sex with Men, generally young adults, white-skinned and with higher schooling levels [15,16,17]. High schooling levels are found among users in Germany, Belgium and Brazil [16,18,19]. However, not all studies included race/skin colour or gender identity and affective-sexual orientation as social variables investigated, restricting the analysis to the sex declared at the participants’ birth and sexual partnerships.

Although the Brazilian Unified Health System (SUS) was established to ensure universal access to healthcare, Black individuals continue to face substantial barriers within this system. Research indicates that, although this population has greater coverage through the Family Health Strategy, they are less likely to have private health insurance and more frequently report difficulties accessing healthcare services compared to white individuals, highlighting persistent inequalities [20,21]. These obstacles are intensified by structural and institutional racism, which not only limits access but also compromises the quality of care provided, ultimately leading to worse health outcomes and reduced quality of life for Black communities [22,23].

In addition, in Brazil and other countries, there is a strong relationship between gender identity and poverty. Women and transgender individuals often face significant economic disadvantages compared to cisgender men, due to factors such as gender-based discrimination, unequal access to education and employment opportunities and disparities in poverty rates between female- and male-headed households [24,25]. Studies that examine the concept of intersectionality help to understand the overlapping systems of oppression and privilege, such as race, gender, social class and sexual orientation, which intersect and influence each other in ways that shape people’s identities and lived experiences. These intersecting contexts create complex forms of inequality and discrimination, while also giving rise to resilience and coping strategies [26,27].

Studies on HIV vulnerability have largely focused on sexual practices [28] and reinforce that cisgender and gay men access HIV PrEP more than other population groups, as shown by the results found in this research. It should be noted that most studies focus on sexual practices involving lesbians and gay men in an attempt to identify differences and similarities [29,30] and end up reinforcing the idea that they are the ones most at risk of transmission, reproducing the blame for AIDS spread. However, LGBT+ people have historically sought more services for STI prevention and treatment, as shown by the data herein reported [31].

This understanding needs to be particularly incorporated among health professionals, who find it difficult to approach sexual practices, especially those of LGBT+ people, who are invariably perceived as deviating from the social norm and categorised in a dichotomous way between heterosexuals and those with some other affective-sexual orientation [32]. This weakness in the professional-patient relationship means that the approach to sexual health is based on stigmas and stereotypes, neglecting the fact that heterosexuals’ sexual practices are also vulnerable to STIs.

Despite the countless efforts aimed at HIV prevention, it is necessary to increase the promotion of HIV testing and awareness about PrEP in different social groups, especially those not representative of the LGBT+ community, as health education efforts are not adequately reaching considerable and different groups of people at risk of HIV infection [33]. Studies in Brazil emphasize that prejudice and discrimination against people living with HIV (serophobia) can discourage testing and the use of preventive strategies such as PrEP, which can result in late diagnoses and reinforce social isolation [34,35].

Given the advances in treatment and prevention, the meaning of HIV and AIDS has changed among people [32], influencing adherence to condom use in sexual relations, as seen in the findings of this study. Once considered a lethal diagnosis, HIV infection has come to be seen as a chronic, transmissible and manageable disease with the discovery of antiretroviral therapies and PrEP as a highly effective tool for HIV prevention [32]. A research study carried out in Australia found a large increase in people in stable relationships and sexual practices with consenting or non-consenting casual partners. From a sample of 3764 gay and bisexual men, 40% reported non-consensual sex with casual partners, especially by participants on PrEP [36].

In this study, the bisexual participants reported syphilis more than homosexuals. However, there is scarcity of research focused on knowledge about sexual desire, orientation and identity expressions in understudied groups within the LGBT+ community, as they are mostly focused on gays and lesbians [30].

This broader understanding of the preventive context may even contribute to the apparent contradiction identified between the descriptive data and the univariate analysis in this study. Although gay men frequently reported having condomless sex, the adjusted results indicated a lower chance of this practice when compared to heterosexuals. This finding may reflect specific strategies adopted by this group, such as consistent PrEP use, serological agreements [37] and incorporation of the U = U concept [38], demonstrating certain reconfiguration of sexual practices in the face of prevention medicalization.

Also in the field of social vulnerabilities, there was a higher chance of exchanging sex for money among cisgender women and, especially, in transgender ones. This result is supported by the literature and is related to the low schooling levels and to exclusion from the formal labour market frequently experienced by these population groups. In a study conducted in the United States of America (USA) with 30,327 users of a sexual health clinic, 34% of the people who reported exchanging sex for money were cisgender women, while 5% were transgender individuals [39]. In another study, carried out in Jamaica, 51.82% of the transgender women reported involvement in sex work, with 47.06% indicating participation in paid sex [40]. These conditions are aggravated by the absence of specific public policies aimed at ensuring that people stay in school and enter the job market, which contributes to maintaining programmatic and social vulnerability, exerting a direct influence on the greater susceptibility of certain groups to communicable diseases [41].

The results presented in Table 2 highlight that cisgender and transgender women are disproportionately affected by social and programmatic vulnerability, with significantly higher odds of engaging in transactional sex and reporting crack use compared to cisgender men. These findings emphasize the layered burden faced by these populations, not only due to their gender identity but also due to the intersection of poverty, limited schooling and exposure to stigmatized forms of survival, such as sex work and substance use [42]. The literature suggests that the association between gender identity, social exclusion and risk behavior is not merely behavioral, but reflects deeply rooted structural inequalities that condition how individuals access prevention services and navigate daily life [42,43].

Additionally, the data underscore a significant: Despite bisexual individuals reporting fewer risk behaviors such as transactional sex and condomless sex, they showed higher odds of a syphilis diagnosis when compared to heterosexuals. This paradox highlights a potential gap in care, whereby bisexual individuals may not be adequately reached by testing, follow-up and treatment strategies tailored to their specific needs. Existing stigma surrounding bisexuality, both in heterosexual and LGBTQIA+ contexts, can contribute to invisibility and neglect within health services, resulting in underdiagnosis, delayed treatment and cumulative exposure to [44] STIs. These results reinforce the importance of intersectional and inclusive approaches to prevention that move beyond identity labels and consider how structural stigma and access to healthcare intersect.

Internationally, it should be noted that although Brazil has made progress in incorporating PrEP into the SUS, inequalities in access persist. Countries with non-universal systems, such as the United States of America (USA), face even greater obstacles, recently accentuated by the cut in global funding for HIV/AIDS [28]. This retraction might compromise programs in Global South countries, directly impacting key populations such as those in the sample of this study.

In addition, the association between psychoactive substance use and risky sexual practices among gay and bisexual men suggests the presence of chemsex—a practice involving multiple sexual partners, drug use and reduced condom use, which is still underdiscussed in Brazilian healthcare services [45]. Chemsex usually encompasses substances such as methamphetamine, gamma-hydroxybutyric acid (GHB), mephedrone and cocaine, employed to intensify sexual experiences, which frequently results in prolonged sessions and in more partners. Some studies [46,47] indicate that up to 30% of gay and bisexual men have reported engaging in chemsex in recent months, revealing high rates of this practice in this group. This dynamic is associated with an increase in risky sexual behaviours such as condomless sex, group sex and higher incidence of sexually transmitted infections, especially gonorrhoea and chlamydia.

It is important to note that chemsex is not a barrier to initiating or continuing PrEP. However, a number of studies point to high prevalence of this practice among transgender men who seek counselling for PrEP [45], highlighting the need for services that address the “substance use” and “sexual health” dimensions in this population segment in an integrated manner. The absence of harm reduction policies aimed at this population can aggravate their exposure to STIs, requiring intersectoral actions between the sexual health and mental health areas [48,49].

Although Brazil has advanced in incorporating PrEP into its public health system, significant disparities remain. This study highlights that cisgender, white and more educated individuals are the main users, while transgender women, cisgender women, bisexuals and Black people face greater barriers and vulnerabilities. These gaps are linked to structural racism, gender inequality, stigma and limited discussions on topics such as chemsex. Expanding access to PrEP requires inclusive, culturally sensitive approaches that address these intersectional challenges.

This study has some limitations. As the data were self-reported and extracted from clinical records, sensitive information may have been omitted. In addition, its cross-sectional design does not allow causal relationships to be established between the variables. The absence of data on income, area of residence and support networks also limits fully understanding the vulnerabilities involved. Although the study was carried out in one of the main specialized HIV/AIDS services in the city of São Paulo with over 40 years of experience, the fact that it was conducted in a single service may restrict generalization of the results to other contexts and populations. Despite these limitations, the findings herein presented contribute to broadening the debate on access to PrEP, the diversity of experiences among its users and the challenges in devising public policies that are sensitive to the multiple dimensions of vulnerability to HIV.

## 5. Conclusions

The results of this study demonstrate that sexual orientation and gender identity are associated with different vulnerability patterns among HIV pre-exposure prophylaxis users, especially regarding psychoactive substance use, risky sexual practices and incidence of sexually transmitted infections. Transgender and bisexual individuals showed higher exposure to risk contexts, highlighting the need to broaden the focus on these groups’ specificities within combined prevention strategies.

In this sense, it is indispensable that public policies and health services incorporate the intersectionality perspective in the care provided and in formulating preventive actions. Strategies adapted to these individuals’ realities, which take into account their social experiences and identity markers and exclusion contexts, are fundamental to enhancing PrEP effectiveness, reducing health inequalities and promoting equitable, comprehensive and humanized access to HIV prevention.

## Figures and Tables

**Table 1 healthcare-13-01841-t001:** Sociodemographic characteristics, sexual practices and STI-related outcomes among users who initiated HIV pre-exposure prophylaxis, by gender identity and sexual orientation. Brazil, 2025 (N = 736).

Variables	Gender Identity			*p*-Value	Sexual Orientation			*p*-Value
	Cisgender Men N/%	Cisgender Women N/%	Transgender Women N/%		Homosexuals/ Gays/Lesbians N/%	Bisexuals N/%	Heterosexuals N/%	
Race/Skin colour				<0.001 ^b^				0.002 ^a^
Asian	17/2.5	0/0.0	0/0.0		15/2.4	2/2.6	0/0.0	
White	494/71.3	11/52.4	6/27.3		449/71.6	47/61.8	15/45.5	
Indigenous	4/0.6	0/0.0	1/4.6		4/0.6	0/0.0	1/3.0	
Brown	132/19.0	9/42.9	14/63.6		120/19.1	18/23.7	17/51.5	
Black	46/6.6	1/4.8	1/4.5		39/6.2	9/11.8	0/0.0	
Total	693/100	21/100	22/100		623/100	76/100	33/100	
**Schooling**				<0.001 ^b^				<0.001 ^b^
No studies	1/0.1	0/0.0	0/0.0		0/0.0	1/1.3	0/0.0	
4–7 years	6/0.9	2/9.5	5/22.7		6/1.0	2/2.6	5/15.2	
8–11 years	91/13.1	5/23.8	9/40.9		75/12.0	21/27.6	9/27.3	
12+ years	595/85.9	14/66.7	8/36.4		546/87.1	52/68.4	19/57.6	
Total	693/100	21/100	22/100		627/100	76/100	33/100	
Sex with people living with HIV				0.456 ^a^				0.742 ^a^
No	196/28.5	5/23.8	9/40.9		177/28.4	22/29.3	11/33.3	
Don’t know	322/46.8	11/52.4	11/50.0		290/46.6	39/52.0	15/45.5	
Yes	170/24.7	5/23.8	2/9.1		156/25.0	14/18.7	7/21.2	
Total	688/100	21/100	22/100		623/100	75/100	33/100	
Exchanging sex for money				0.001 ^a^				0.001 ^a^
No	595/86.1	8/38.1	6/27.3		550/88.0	47/61.8	12/36.4	
Yes	96/13.9	13/61.9	16/72.7		75/12.0	29/38.2	21/63.6	
Total	691/100	21/100	22/100		625/100	76/100	33/100	
Condomless sex				0.017 ^b^				0.413 ^b^
No	7/1.1	2/10.0	1/5.6		8/1.4	1/1.5	1/3.6	
Yes	605/98.9	18/90.0	17/94.4		546/98.6	67/98.5	27/96.4	
Total	612/100	20/100	18/100		554/100	68/100	28/100	
Syphilis diagnosis				0.336 ^b^				0.988 ^a^
No	64/49.6	2/50.0	0/0.0		54/48.2	10/50.0	2/50.0	
Yes	65/50.4	2/50.0	3/100.0		58/51.8	10/50.0	2/50.0	
Total	129/100	4/100	3/100		112/100	20/100	4/100	
Rectal gonorrhoea and/or chlamydia				0.795 ^b^				0.648 ^b^
No	100/77.5	4/100.0	3/100.0		88/78.6	15/75.0	4/100.0	
Yes	29/22.5	0/0.0	0/0.0		24/21.4	5/25.0	0/0.0	
Total	129/100	4/100)	3/100		112/100	20/100	4/100	
Any substance use				0.838 ^a^				0.716 ^a^
Total	680/100	20/100	22/100		615/100	75/100	32/100	
No	304/44.7	10/50.0	9/40.9		279/45.4	31/41.3	13/40.6	
Yes	376/55.3	10/50.0	13/59.1		336/54.6	44/58.7	19/59.4	
Alcohol	18/58.1	0/--	1/100.0	1.000 ^b^	17/60.7	1/33.3	1/100.0	0.737 ^b^
Poppers	106/28.2	1/10.0	1/7.7	0.124 ^a^	97/28.9	10/22.7	1/5.3	0.063 ^a^
Cocaine	106/28.2	3/30.0	7/53.9	0.135 ^a^	96/28.6	10/22.7	10/52.6	0.050 ^a^
Crack	4/1.1	1/10.0	1/7.7	0.041 ^b^	4/1.2	0/0.0	2/10.5	0.048 ^b^
Cannabis	233/62.0	8/80.0	9/69.2	0.450 ^a^	211/62.8	26/59.1	13/68.4	0.775 ^a^
Club drugs	164/43.6	1/10.0	4/30.8	0.073 ^a^	143/42.6	20/45.5	6/31.6	0.583 ^a^
Erectile stimulants	159/42.3	1/10.0	0/0.0	0.001 ^b^	140/41.7	20/45.5	0/0.0	0.001 ^b^
Solvents	33/8.8	0/0.0	3/23.1	0.150 ^b^	25/7.4	8/18.2	3/15.8	0.038 ^a^
Age (years old): Mean 34.9 (SD = 9.1)								

^a^ Pearson’s Chi-square test; ^b^ Fisher’s Exact test. SD: Standard Deviation.

**Table 2 healthcare-13-01841-t002:** Association between gender identity/sexual orientation and incidence of Sexually Transmitted Infections STIs), psychoactive substance use and sexual practices among the users who initiated HIV pre-exposure prophylaxis. Brazil, 2025. N = 736.

Variables	Gender Identity			Sexual Orientation		
	Cisgender Men (Ref)	Cisgender Women (OR 95%CI)	Transgender Women (OR 95%CI)	Heterosexuals (Ref)	Homosexuals/ Gays/Lesbians (OR 95%CI)	Bisexuals (OR 95%CI)
Exchanging sex for money	Ref	9.8 (3.5–27.3)	15.6 (5.3–46.1)	Ref	0.1 (0.0–0.2)	0.4 (0.2–1.0)
Condomless sex	Ref	0.1 (0.0–0.5)	0.1 (0.0–1.2)	Ref	0.1 (0.0–0.6)	0.1 (0.0–2.7)
Syphilis diagnosis	Ref	1.0 (0.10–9.4)	6.9 (0.1–339)	Ref	29.0 (1.1–762)	34.6 (1.2–960)
Crack use	Ref	13.1 (1.31–130)	9.9 (1.0–94.7)	-	-	-

Ref: Reference group (Cisgender men; heterosexuals). OR: Odds Ratio, estimated by logistic regression. 95%CI: 95% Confidence Interval. OR values with confidence intervals not including 1 are considered statistically significant.

## Data Availability

The data are available upon reasonable request.

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
