# Peer review of "Sexual Orientation and Gender Identity Associated with Sexual Practices, Psychoactive Substance Use and Sexually Transmitted Infections Among HIV PrEP Users"

_healthcare, 2025, doi:10.3390/healthcare13151841_

Round 1

Reviewer 1 Report

Comments and Suggestions for Authors

I think this is an interesting and important paper. However, there are a lot of generalities about healthcare workers providing care for the LGBT+ community that I think need to be tempered and/or qualified. I think more background on the healthcare system in Brazil would also be helpful. Your discussion section needs to be reworked. There are a lot of generalizations that are not supported by the data provided.

Page 2, line 44 - I would recommend against using the term "transvestite" as it has very negative connotations and is not further used in the manuscript

Lines 50-55 - This paragraph is convoluted and difficult to follow

Lines 73-75 - The aim statement needs to be stated more clearly. 

Lines 91-94 - I'm confused as to why return visits are discussed when there is no data presented on compliance or subsequent seroconversion.

Results - Please provide an N for the overall groups with the percentages, in the text and Table 1

You mention age in the first paragraph of the results but then not again. Is there a difference in the ages between the groups. 

Table 2 - this table is a little overwhelming; I'm wondering if you could either combine or remove the sores/blisters/discharge since there isn't any statistical significance

Lines 163-165 - Reword this sentence for improved readability

Lines 197-199 - I think you're trying to say that your sample is biased because it represents those who are seeking care with PrEP rather than all patients at risk for HIV. However, this sentence is very hard to interpret and does not convey that message clearly. I'd like to hear more about healthcare in Brazil and barriers for black and brown people.

Lines 201-203 - need to clarify that this systematic review only looked at care in the US.

Lines 213-236 - while important, I don't feel like these paragraphs have much to do with your study

Lines 269-271 - I don't think you can say this. It certainly is not supported by your findings.

Lines 271-272 - If you're talking about the literature, then you need to site supporting sources.

Lines 308-310 - I don't understand this statement. Does it mean that it is debated? Or that providers don't think it exists?

I think your final paragraph of the discussion is good and pretty succinctly summarizes what you are trying to say. However, this messaging gets lost in the rest of your discussion. 

Comments on the Quality of English Language

Some of your language is unnecessarily complex and would greatly benefit from a native English speaker editing your manuscript.

Author Response

We thank the reviewer for the observation. We have carefully revised the manuscript to improve clarity and language fluency. Should the manuscript be accepted, we are committed to submitting it for professional English editing if requested.

Reviewer 2 Report

Comments and Suggestions for Authors

This is a well written manuscript that addresses important topic among members of key population individuals.

Minor comments, mainly in the methods and results section

  1. The analysis does not show the association of participants characteristics for tables 1-3. Until when the reader reads the footnote of those table. There is s need to include that Chi-square or Fisher’s exact tests where use where appropriate in the analysis section
  2. Did you have any missing values? If you did, how were these missing values being handled? This information needs to be incorporated in the analysis section
  3. Only univariate logistic regression models were conducted. Is there a reason as to why multivariable analysis was not done? Multivariable would distinguish confounders of these associations
  4. The odds ratios were interpreted as risk ratio by using word such as more likely or less likely. These words are used for the interpretations of risk ratio/ rate ratios and hazard ratios. For odds ratio, it is appropriate to consider stating this group has higher odds/lower odds of this outcome compared to that group. Or Increased/reduced odds.

Author Response

Please see the attached file for the detailed point-by-point response to the reviewer.
